# Ferroptosis and Its Potential Role in Glioma: From Molecular Mechanisms to Therapeutic Opportunities

**DOI:** 10.3390/antiox11112123

**Published:** 2022-10-28

**Authors:** Yusong Luo, Guopeng Tian, Xiang Fang, Shengwei Bai, Guoqiang Yuan, Yawen Pan

**Affiliations:** 1Department of Neurosurgery, Lanzhou University Second Hospital, Lanzhou 730030, China; 2Key Laboratory of Neurology of Gansu Province, Lanzhou 730030, China; 3The Second Clinical Medical School, Lanzhou University, Lanzhou 730030, China

**Keywords:** glioma, ferroptosis, lipid peroxidation, molecular mechanisms, treatment

## Abstract

Glioma is the most common intracranial malignant tumor, and the current main standard treatment option is a combination of tumor surgical resection, chemotherapy and radiotherapy. Due to the terribly poor five-year survival rate of patients with gliomas and the high recurrence rate of gliomas, some new and efficient therapeutic strategies are expected. Recently, ferroptosis, as a new form of cell death, has played a significant role in the treatment of gliomas. Specifically, studies have revealed key processes of ferroptosis, including iron overload in cells, occurrence of lipid peroxidation, inactivation of cysteine/glutathione antiporter system Xc^−^ (xCT) and glutathione peroxidase 4 (GPX4). In the present review, we summarized the molecular mechanisms of ferroptosis and introduced the application and challenges of ferroptosis in the development and treatment of gliomas. Moreover, we highlighted the therapeutic opportunities of manipulating ferroptosis to improve glioma treatments, which may improve the clinical outcome.

## 1. Introduction

Glioma is the most common primary malignant tumor of the brain, accounting for approximately 50–60% of the central nervous system (CNS) tumors [1] and approximately 81% of intracranial malignancies [2,3]. Patients with gliomas have significantly higher recurrence rates than those with other tumors of the CNS [4]. Gliomas have been classified by the World Health Organization (WHO) grading system into four grades, where gliomas of grade 1 and grade 2 indicate low-grade gliomas, and gliomas of grade 3 and grade 4 reveal high-grade gliomas [5]. The median overall survival (OS) time of low-grade glioma patients is approximately 11.6 years [6]. However, patients with grade 3 glioma have a median OS time of approximately three years, and the median OS time of grade 4 glioma patients is approximately 15 months [7]. While current clinical treatments for glioma consist of surgical resection, radiotherapy, chemotherapy, novel molecular targeted therapy and immunotherapy [8], these treatments have not brought desirable benefits to patients, and the prognosis of patients remains extremely poor [9,10]. Therefore, there is a great need to develop new therapeutic strategies, including novel therapeutic targets inhibiting glioma cells, to improve OS time and the quality of life for these patients. The common deaths of different cells in the body include necrosis, apoptosis, autophagy and pyroptosis [11]. Recently, ferroptosis, as a new nonapoptotic cell death pattern resulting from iron-dependent lipid peroxidation injury, has attracted more attention [12,13,14].

Ferroptosis is a new type of programmed cell death triggered by cell membrane damage arising from these processes, including intracellular iron accumulation, production of reactive oxygen species (ROS), lipid peroxidation, failure activity of glutathione peroxidase (GPX) and xCT [15,16]. Cells undergoing ferroptosis not only have changes in cell composition, but also in cell morphology. When cells undergo ferroptosis, although the morphology of the nucleus does not change significantly, the morphology of mitochondria shows increased bilayer membrane density, reduction or disappearance of cristae, and reduced volume [16,17]. Many recent studies have shown that significant progress has been made on the impacts of ferroptosis on glioma. Ferroptosis inducers, as compounds from plants and others, have certain effects on the treatment of glioma by affecting ferroptosis processes.

Herein, we will mainly focused on the essential molecular mechanisms of ferroptosis, as well as the potential impact of ferroptosis on glioma growth and treatment. We also provided an overview of the challenges of ferroptosis in glioma therapy and discussed the therapeutic opportunities of manipulating ferroptosis to improve treatment.

## 2. Molecular Mechanisms of Ferroptosis

J.M. Gutteridge found in 1984 that iron salts could induce lipid peroxidation by breaking down lipid peroxides into alkoxyl and peroxyl radicals, and iron complexed with ethylene diamine tetraacetic acid (EDTA) could also initiate lipid peroxidation by reacting with hydrogen peroxide (H_2_O_2_) to form hydroxyl radicals (^•^OH), which may lay the foundation for iron-dependent cell death [18]. In addition, it has been reported that exogenous glutamate could induce cell death by inhibiting cystine uptake through xCT to lead to decreased glutathione production, and a unique programmed cell death pathway called oxytosis, which was dependent on oxidative stress and ROS production and was introduced [19]. This laid the prior groundwork for the discovery and proposal of ferroptosis. Ferroptosis was defined as a new form of programmed cell death by Brent R. Stockwell in 2012 that expresses the process of iron-dependent cell death in cancer cells [13]. Molecular mechanisms of ferroptosis differ from other major forms of regulated cell death (RCD) (Table 1). The main biochemical processes of ferroptosis consist of excess iron and accumulation of ROS in cells, lipid peroxidation, inactivation of xCT and depletion of glutathione and lipid repair enzyme [20,21,22].

### 2.1. Iron Metabolism

Iron is a basic trace element for various cells to carry out various biological functions. Dietary iron comes in many forms but is typically classified as either non-heme or heme iron (Fe^2+^ complexed with protoporphyrin IX). Non-heme dietary iron exists largely as ferric salts, which are reduced back to Fe^2+^ by iron reductase in the intestine. Fe^2+^ enters intestinal epithelial cells (IECs) via the brush-border transporter divalent metal transporter 1 (DMT1) and exits through ferroportin 1 (FPN1) in the basolateral membranes [23,24]. Fe^2+^ is oxidized to Fe^3+^ by ceruloplasmin (CP) and hephaestin (HP), and Fe^3+^ combines with transferrin (Tf) to be transported in the blood [25]. Tf-Fe^3+^ attaches to the transferrin receptor (TfR) on the cell membrane and then internalizes to the cell as endosomes [26,27]. Fe^3+^ is released and reduced to Fe^2+^ by 6-transmembrane epithelial antigen of the prostate 3 (STEAP3) in a variety of cells; then, Fe^2+^ enters the cytoplasm via DMT1 on the endosomal membrane [28,29]. Heme iron is a part of the hemoproteins hemoglobin and myoglobin. However, the molecular mechanism of heme iron absorption is still unclear. There is some evidence that ingested heme iron is decomposed by heme oxygenase in intestinal cells, thus releasing free ferric iron. A large amount of Fe^2+^ accumulates in the cytoplasm to form a labile iron pool, and the metabolic activity of Fe^2+^ has a vital impact on various biological functions, such as ferroptosis [30,31]. Intracellular iron overload, with H_2_O_2_, triggers the Fenton reaction, inducing the formation of ROS, such as ^•^OH, which cause lipid peroxidation to provoke ferroptosis [32,33,34] (Figure 1). Iron responsive element binding protein 2 (IREB2), as a significant regulator of iron metabolism, may develop sensitivity to ferroptosis [35,36,37]. Meanwhile, autophagy can regulate the iron pool by affecting the recruitment of ferritin to autophagosomes for lysosomal degradation to release free iron [38,39,40]. For example, ferritinophagy directly recognizes the ferritin heavy chain 1 (FTH1) by the cargo receptor nuclear receptor coactivator 4 (NCOA4) and then releases iron by transporting the ferritin complex to autophagosomes for lysosomal degradation [41]. Conversely, reduced intracellular Fe^2+^ levels could impede the process of ferroptosis. For instance, erastin-induced ferroptosis was weakened by decreased intracellular Fe^2+^ because of the knockout of autophagy-related 5 (ATG5) or autophagy-related 7 (ATG7) [42]. Thus, the metabolism of iron plays a vital role in ferroptosis.

### 2.2. Lipid Metabolism

Lipid peroxidation, a hallmark feature of ferroptosis, is the ultimate executor of ferroptosis. ROS generated by the Fenton reaction interact with polyunsaturated fatty acids (PUFAs) on cellular or organelle membranes to generate toxic phospholipid hydroperoxides (PLOOHs), thereby inducing ferroptosis [43,44,45]. Research has shown that some factors, such as acyl–coenzyme A synthetase long-chain family 4 (ACSL4), lysophosphatidylcholine acyltransferase 3 (LPCAT3) and lipoxygenases (LOXs), participate in the production of lipid peroxidation [46,47,48]. ACSL4 (as a required lipid metabolism enzyme) and LPCAT3 (as a class of key enzymes catalyzing the reacylation of lysophospholipids to phospholipids) activate free long-chain polyunsaturated fatty acids, promote lysophosphatidylcholine (LPC) conversion into lecithin, mediate the synthesis of oxidized cell membrane phospholipids, and subsequently regulate ferroptosis development [49,50]. Meanwhile, ACSL4 esterifies arachidonic acid (AA) into acyl-coenzyme A (acyl-CoA) for the biosynthesis of PUFAs, which plays a key role in lipid peroxidation and ferroptosis [51]. LOXs, pivotal regulators of ferroptosis, may have a vital effect on the initiation of ferroptosis by promoting lipid autoxidation and predicting ferroptosis sensitivity [47,52]. Therefore, lipid peroxidation in ferroptosis executes cell death by the destruction of the lipid bilayer on cellular or organelle membranes (Figure 2a).

### 2.3. The xCT and GPX4

Environmental pressure (such as high temperature and hypoxia) can cause iron reaction, so the cell also needs to establish an appropriate mechanism of defense ferroptosis. The most classic defense way of ferroptosis is the antioxidant axis formed by xCT, glutathione (GSH), and GPX4. The xCT, as a transmembrane protein, consists of light-chain solute carrier family 7 member 11 (SLC7A11) and heavy-chain solute carrier family 3 member 2 (SLC3A2, CD98hc or 4F2hc). SLC7A11, which is a main functional subunit of xCT, aims to regulate extracellular cysteine (Cys) into cells and intracellular glutamic acid (Glu) out of cells, and SLC3A2, as an important subunit, maintains the stability of xCT by anchoring and stabilizing SLC7A11 [53,54]. Then, Cys generates reduced GSH with Glu and glycine (Gly) under the catalysis of glutamate cysteine ligase (GCL) and glutamylcysteine synthetase (GCS) [55,56]. Beclin 1 suppresses xCT activity to promote ferroptosis by adhering to SLC7A11 directly [57]. GPX4, as a key enzyme in ferroptosis, reduces PLOOH to nontoxic phospholipid alcohols (PLOHs) in membranes with GSH to prevent ferroptosis [58]. The inhibitor 2-cyano-3,12-dioxooleana-1,9 (11)-dien-28-oic acid (CDDO) prevents the specific degradation of GPX4 via chaperone-mediated autophagy (CMA) by affecting the interaction between heat shock protein 90 (HSP90) and lysosomes, which inhibits the ferroptosis of cells [59]. However, the inhibition of CMA is relieved by inhibiting the mammalian target of rapamycin (MTOR) pathway [60], which may involve the degradation of GPX4 to promote ferroptosis. Therefore, xCT, GPX4 and GSH could be regulated and have an important effect on ferroptosis (Figure 2b).

### 2.4. FSP1 and DHODH

In addition to the classic GPX4 defense pathway, recent studies have identified ferroptosis suppressor protein 1 (FSP1) and dihydroorotate dehydrogenase (DHODH) independent of the GPX4 signaling pathway. They all involve ubiquinone, a metabolite molecule that exists in both chemically reduced and oxidized states. Ubiquinone (or CoQ) is a lipid that functions on cell membranes and mitochondrial membranes. FSP1 in the cell membrane inhibits ferroptosis by reducing ubiquitin to ubiquinol (CoQH2), which acts as a free radical trapping antioxidant to prevent lipid peroxidation at the cell membrane [61,62]. Similar to the FSP1 system mechanism, DHODH-mediated regulation of panthenol production is an effective system that is specifically designed to alleviate lipid peroxidation in the mitochondria. Mitochondria produce a large amount of ROS in the electron transport chain located in the inner membrane during oxidative phosphorylation. Lipid peroxidation occurs when the mitochondrial antioxidant system is damaged and unable to remove ROS. DHODH is a flavin-dependent enzyme located in the mitochondrial inner membrane and its main function is to catalyze the fourth step of the pyrimidine biosynthesis pathway [63]. The oxidation of dihydroorotate (DHO) to orotate (OA) simultaneously transfers electrons to the ubiquitin in the inner mitochondrial membrane for its reduction to ubiquinol. Gan’s team found that when cells were treated with GPX4 inhibitors, such as RSL3, metabolomic analysis revealed a significant decrease in N-aminoformyl-aspartate (C-Asp), increase in Uridine and increase in the synthesis of uridine 5′-monophosphate (UMP) [64]. This suggests a possible relationship between ferroptosis and pyrimidine nucleotide synthesis. By supplementing intermediate metabolites for pyrimidine synthesis, the authors found that dihydroorotic acid inhibited RSL3-induced ferroptosis, whereas orotic acid made cells more sensitive to RSL3. Since DHO and OA are substrates and products of DHODH, respectively, this further confirms that DHODH may be involved in the regulation of ferroptosis. Interestingly, further studies have revealed the use of the DHODH inhibitor Brequinar (BQR) to induce ferroptosis in GPX4 low-expression cells, while the high expression of GPX4, BQR treatment significantly increased the sensitivity of cells to ferroptosis inducers. As is known, most GBM cell lines have higher expression levels of DHODH and GPX4 compared to normal human astrocyte cytoplasm (NHA) [65]. The research also confirms that, in the solid tumor with a high expression of GPX4, the combination of ferroptosis inducer sulfasalazine and DHODH inhibitor can have good therapeutic effect. This finding provides a new strategy for how to target ferroptosis in glioma therapy.

In general, there are at least three types of iron-death defense systems in cells based on different subcellular localization: GPX4 in the cytosol and mitochondria, FSP1 in the cell membrane, and DHODH in mitochondria.

## 3. Targeting Ferroptosis to Treat Glioma

### 3.1. Metabolic Pathway

Iron in the brain plays a crucial role in maintaining proper functioning of the central nervous system through its participation in many cellular activities, such as myelination, neurotransmitter synthesis, and energy production. The maintenance of this homeostasis depends on the function of the blood–brain barrier (BBB), which is composed of brain microvascular endothelial cells (BMVEC), astrocytes, microglia and pericytes. It has long been thought that the development of the BBB leads to a reduction in iron absorption in infancy. However, some evidence has revealed that iron levels in the brain increase with ageing [66,67]. Additionally, the mechanism of iron uptake into BMVEC is thought to be primarily via the Tf/TfR1 pathway. Iron was first uptake into BMVEC from blood by TfR1-mediated endocytosis. Then, the action of an H+-ATPase, the membrane of endocytosomes reduced the pH of the endocytosomes, resulting in the dissociation of Fe^3+^ from Tf and their reduction to Fe^2+^, which will cross the endosomal membrane by a process mediated by DMT1. When the PH rises to 7.4, the non-iron-bound Tf (apo-Tf) and TfR1 return to the luminal membrane where apo-Tf-TfR1 is released into the blood for the next round of iron uptake [68]. Interestingly, after iron was released into the brain interstitial fluid, it was unevenly distributed among different cell types in different regions of the brain [69]. Almost all iron transport-related proteins are expressed in glial cells, but not in neurons.

As is known, iron is a cofactor for many enzymes, including ribonucleic acid reductase (RR), which is an enzyme involved in DNA synthesis. To maintain proliferation, GBM cells need to increase iron uptake, thereby regulating the expression of proteins involved in iron uptake. Recently, studies have reported that there are higher free iron levels in glioma than in other brain tumors, such as meningioma cells and glioblastoma cancer stem cells [70]. Iron-related gene expression in gliomas, such as TfR1 and TfR2, is different from that in other brain tumors and normal human brain tissue [71,72]. TfR levels in glioma sample tissues appear higher than those in meningiomas and other brain tumors in general, which may be correlated with the high levels of iron in gliomas [73]. In addition, the high expression of TfR2 not only promotes glioma cell proliferation, but also contributes to the better sensitivity to temozolomide. The proliferation of glioma cells is attributed to TfR2, which could be localized in lipid rafts and stimulate the ERK1/ERK2 phosphorylation by combining with Tf, but the mechanism of TfR2-induced glioma hypersensitivity to temozolomide remains unclear [74]. Therefore, the effect of TfR2 on glioma is still controversial, and more reports are needed to verify this. Several recent studies have indicated that iron homeostasis and ferroptosis are also affected by iron-sulfur cluster (ISC) proteins. Loss of ISC could induce ferroptosis by initiating iron-starvation responses to lead to iron overload in tumor cells. This mechanism is that ISC synthesis inhibition could activate the iron regulatory protein (IRP); IRP increases TfR levels and reduces FPN1 production by binding to target mRNAs, which promotes intracellular excessive iron accumulation [75]. However, whether ISC has a similar role in glioma requires more evidence.

DMT1 may be related to increased iron levels in glioma cells and is currently a molecular marker in neurodegenerative diseases [76]. In an experimental rat model with C6 glioma cells, propofol inhibits DMT1 expression, tumor cell proliferation and eventually decreased glioma weight [77]. Additionally, this tumor suppressive effect was further found to be associated with a significant reduction in the GSH and ROS. However, a study showed that temozolomide (TMZ) may suppress tumor growth by inducing ferroptosis by targeting DMT1 expression in glioblastoma cells [78]. These results suggest that DMT1 may affect glioma proliferation by regulating ferroptosis and ROS levels and has been investigated as a potential therapeutic target. While the STEAP3 protein plays a vital role in other processes, such as affecting the inflammatory response by regulating the Toll-like receptor 4-mediated macrophage production of chemoattractant protein-5, interferon-beta and interferon-induced protein-10 [79,80], it could also have an essential impact on ferroptosis by reducing Fe^3+^ to Fe^2+^ [81]. The expression of STEAP3 in glioma cells is higher than that in normal brain tissues, which could be regarded as a potential prognostic marker and reduce the overall survival of patients with glioma [82,83]. In addition, STEAP3 not only regulates ferroptosis by enhancing TfR expression and inducing mesenchymal transition, but also has a direct influence on glioma cell proliferation, invasion, and sphere formation in vitro and on glioma growth in vivo [84]. Poly(C)-binding protein 2 (PCBP2), as a significant factor in iron metabolism and posttranscriptional and translational regulation, possibly affects the process of ferroptosis. While PCBP2 is upregulated in glioma tissues and cell lines, the development and proliferation of glioma are suppressed when it is knocked down or when its inhibitor microRNA-214 is applied [85]. The higher levels of ferritin detected in the serum of patients with tumors possibly predict that the prognosis of these patients will deteriorate more, which indicates that iron metabolism plays a necessary role in the progression and therapy of tumors. Ferritins in the serum and cerebrospinal fluid of patients with gliomas, which could come from glioma cells, were higher than those of patients without gliomas [86]. Significant evidence has suggested that nuclear factor erythroid-related factor 2 (NRF2) acts as a key regulator of antioxidant responses, which favors cancer cell growth and leads to increased drug resistance in tumor cells [87,88,89]. NRF2 mainly targets heme oxygenase-1 (HO-1) to reduce the levels of ROS and degrade prooxidants [90]. Research has shown that neuronal precursor cell-expressed developmentally downregulated 4-1 (NEDD4-1) induces resistance to TMZ treatment in gliomas via activating the AKT/NRF2/HO-1 axis [91]. TMZ also induces ferroptosis by inhibiting the NRF2/HO-1 signaling pathway in gliomas [78]. NRF2/HO-1axis appears to play an important role in glioma therapy. In addition, triptolide and brusatol, as NRF2 inhibitors, suppresses potently IDH1-mutated glioma cells by targeting the NRF2-driven glutathione synthesis pathway to induce lipid peroxidation [92,93]. The upregulated cystathionine-γ-lyase (CSE) in IDH1-mutant astrocytomas promotes cell survival by maintaining GSH to drive antioxidant defense, and whether it is related to the NRF2 needs further verification [94]. Therefore, ferroptosis in gliomas could be associated with the regulation of NRF2.

### 3.2. The xCT Pathway

Cysteine deprivation is an important inducer of ferroptosis and greatly contributes to the ferroptosis in GBM [95,96]. The study conducted by Takeuchi et al., including 40 patients with gliomas, concluded that high levels of xCT could predict a short progression-free survival and a low overall survival [97]. Specifically, the high levels of xCT possibly promote glioma cells to grow and survive by enhancing mitochondrial biogenesis and adenosine tri-phosphate (ATP) generation, as well as by reducing the accumulation of ROS [98]. These findings suggest that we can inhibit system Xc-induces ferroptosis in glioma. Radiation, chemotherapy (such as TMZ) and immunotherapy could lead to the activation of ferroptosis by downregulating the expression of xCT to induce the death of glioma cells [99,100,101]. Some widely used clinical drugs have also been applied to the treatment of gliomas by managing xCT and sequentially regulating ferroptosis. Gao et al. found that ibuprofen could enhance ferroptosis by depleting the expression of xCT and GPX4 to inhibit the growth of glioma cells [102]. Another study found that sulfasalazine could stimulate ferroptosis by inhibiting the activity of xCT and sequentially decreasing the formation of GSH strengthened the effect of radiation therapy to increase the overall survival of mice [103]. In addition, tumor suppressor P53 is a frequently mutated gene in various cancers, including glioma. P53 suppresses glioma growth by the induction of ferroptosis [104]. Notably, P53 possibly inhibits the activity of xCT by directly depleting the level of SLC7A11, thus promoting the ability of ferroptosis to suppress the growth of glioma cells [105]. However, an interesting phenomenon is that, in glucose deprivation environments, the treatment of epidermal growth factor will upregulate xCT in glioma cell lines, leading to tumor death [106,107,108,109].

In summary, xCT could play a dual role in the development and treatment of gliomas, so further studies are needed to express the practical effect of xCT in gliomas.

### 3.3. GPX4 Expression

Recently, a study revealed that when GPX4 is knocked down or reacts with its inhibitors, ferroptosis is activated to induce the death of glioma cells by accumulating lipid peroxides to damage the cell membrane and organelle membrane [110]. Some ferroptosis inducers, such as plumbagin, triggers ferroptosis by inducing GPX4 degradation via the lysosome pathway and inhibiting glioma growth [111]. In addition, it has been demonstrated that certain traditional Chinese herbs induce ferroptosis in glioma. For instance, capsaicin, as a potential anticancer ferroptosis inducer, suppresses the proliferative effects of glioma cells by increasing ACSL4 levels and decreasing GPX4 levels to induce ferroptosis [112]. Dihydrotanshinone I (DHI), which boosted ferroptosis by decreasing the expression level of GPX4 and increasing that of ACSL4, inhibited the growth and proliferation of glioma cells [113]. The curcumin analog ALZ003 inhibited the development and growth of glioma cells and enhanced their sensitivity to temozolomide treatment by promoting the androgen receptor (AR) ubiquitination and downregulating GPX4 to highlight ferroptosis in vitro, which improved the survival of experimental rodents in vivo [114]. Additionally, the natural compound artesunate (ART), as an antimalarial drug, was previously demonstrated and ART also exhibited an anti-tumor effect and was a specific inducer of ferroptosis in a number of different types of cancer, including glioma. It inhibits the proliferation of glioma cells in vitro and in vivo by promoting GSH depletion and low GPX4 expression to increase ferroptosis [115]. In another study, dihydroartemisinin (DHA), as an inhibitor, could have the prospect of treating glioma because of its role in promoting apoptosis and autophagy and reducing the invasion ability of glioma cells [116,117]. Specifically, DHA could promote the development of ACSL4 and xCT, but significantly downregulated GPX4 levels, initiating the death of glioma cells by maintaining ferroptosis [118]. Beyond the above stated aspects, there are many new nanomaterials involved. It was reported that the biomimetic nanoparticles (PIOC@CM NPs) increased the level of ROS, depleted GSH upon ultrasonic irradiation and attenuated the activity of GPX4 to kill glioma C6 cells by activating ferroptosis [119]. Iron oxide nanoparticles loaded with paclitaxel (IONP@PTX) not only inhibited the migration and invasion of glioma cells by enhancing ions, ROS and lipid peroxidation, but also promoted the autophagy-dependent ferroptosis pathway by decreasing the levels of GPX4 in vitro [120]. Above all, these findings provide some new drug treatment options for glioma and demonstrate that GPX4 degradation promotes ferroptosis in glioma.

### 3.4. Tumor Immune Microenvironment

As key regulatory components, immune cells and immune-related molecules have been shown to play pivotal roles in the development and treatment of glioma cells. Accumulating evidence indicates that ferroptosis not only promotes tumor cell death, but also affects the tumor immune microenvironment (TIME) [121,122] (Figure 3). As key regulatory components, immune cells and immune-related molecules have been shown to play pivotal roles in the development and treatment of glioma cells. The main tumor-associated macrophages (TAMs) are the M1-polarized subtype with proinflammatory and antitumoral functions and the M2-polarized subtype with anti-inflammatory and protumoral effects [123,124,125]. The M2 subtype plays a vital role in the TIME by excreting extracellular matrix (ECM) components, promoting T-cell anergy and stimulating angiogenesis [126,127]. In addition, TAMs may increase the tumorigenicity and chemoresistance of glioma cells by revising the stromal and blood vessel architecture [128]. Myeloid-derived suppressor cells (MDSCs), as myeloid-derived progenitor cells that accumulate in the TIME of glioma, suppress the proliferation and activity of T cells by releasing inducible nitric oxide synthase, ROS, cyclooxygenase-2 and transforming growth factor-β [129,130]. The regulatory T (Treg) cells accumulating in the TIME of glioma could inhibit immune surveillance and attack by excreting IL-10, IL-35 and TGF-β, which possibly predicts the poor prognosis of patients with gliomas [131,132,133]. Neutrophils are able to not only inhibit the antiangiogenic therapy of other tumors, but also predict the poor prognosis of patients with gliomas [134].

However, the regulation of ferroptosis in the TIME appears to show the treatment limitations of glioma. Due to the reducing supply of oxygen and nutrients, as well as promoting acidosis in the TIME [135,136], glioma cells will survive due to their own tremendous plasticity [137,138,139], but immune cells cannot adapt and lose the effect [140,141]. In addition, the function of immune cells could be inhibited due to the cytokines secreted by glioma cells and immune cells, which may express ferroptosis [142,143]. The Treg cells that impede immune surveillance of tumors, an immunosuppressive subset of CD4+ T cells, are resistant to ferroptosis, which is likely due to GPX4 induction in activated Treg cells [144]. A study has shown that KRAS-G12D can be released into exosomes by pancreatic cancer cells upon ferroptosis and taken up by macrophages via advanced glycation end products (AGEs), which ultimately stimulate tumor growth through the polarization of macrophages to the M2 phenotype. In addition, conditional deletion of Gpx4 in the pancreas of mice promoted mutant Kras-driven tumorigenesis through ferroptotic injury-induced DNA release and subsequent STING-activated inflammation in macrophages [12]. Previous experiments and database analysis showed that the infiltration of Treg cells, neutrophils, and M2-polarized subtype macrophages in the TIME was significantly increased at high levels of ferroptosis [121]. Similarly, MDSCs with immunosuppressive functions exhibit resistance to ferroptosis due to an inhibitory drive on the p53-heme oxygenase 1 (HMOX1) axis mediated by N-acylsphingosine aminohydrolase 2 (ASAH2). In addition, large amounts of lipid peroxides were detected in tumor-derived CD8^+^ T cells, but not in lymph node-derived CD8^+^ T cells, which suggested that ferroptosis may be the metabolic vulnerability of tumor-specific CD8^+^ T cells. In addition, ferroptosis inducers (especially GPX4 inhibitors) may reduce antitumour immunity and promote tumor development by impairing CD8^+^ T cells and follicular helper T cells (Tfhs) [144]. Under these conditions, ferroptosis may promote tumor growth by suppressing antitumour immunity. These advances indicate that ferroptosis has great potential to enhance the immunotherapy in cancer treatment [145,146]. 

## 4. Challenges of Ferroptosis in Glioma

While ferroptosis has a significant effect on the growth and treatment of gliomas, the mechanism needs to be further explored [147]. Most RCD effector molecules are proteases or porins, for example, caspases as well as mixed lineage kinase domain-like protein (MLKL) are involved in apoptosis and necrosis, respectively, and GasderminD participates in pyroptosis [148,149,150]. PLOOH is currently regarded as the ultimate executor of ferroptosis; however, whether effector molecules still exist downstream of PLOOH requires more exploration. [145]. Cytoplasmic GPX4 detoxifying lipid peroxides accumulated at the plasma membrane were unable to inhibit mitochondrial lipid peroxidation-induced ferroptosis in GPX4 knockout cells treated with DHODH inhibitors, which indicated that additional ferroptosis enforcement mechanisms may exist downstream of cytoplasmic lipid peroxidation [144]. The interaction between ferroptosis and other RCDs is also unclear. Some features of ferroptosis compared to other RCDs are not unique; for example, ferroptosis signals such as lipid peroxidation and regulators such as GPX4 and SLC7A11 can also regulate other types of RCDs. Ferroptosis may also modulate the TIME by interacting with other RCDs, thereby affecting the development and treatment of gliomas. In addition, ferroptosis currently has limitations in the diagnosis and treatment of gliomas. Currently, some targets, especially GPX4, ACSL4, P53 and FTH1 [151,152], are regarded as biomarkers, but they are not still a gold standard. Some significant biomarkers that can accurately predict the tumor response to ferroptosis induction are urgently needed, especially those that can be detected directly in patient blood, urine, feces and tumor tissue. It is also not clear what types of patients with gliomas are more sensitive to ferroptosis treatments. Three criteria, including iron levels, gene expression and mutations, can be combined to assess which patients are most likely to benefit from ferroptosis; for example, SLC7A11 inhibitors may be particularly effective against certain types of gliomas that overexpress this target. Finally, ferroptosis plays a dual role in the development and treatment of tumors. Ferroptosis promoted glioma cell death, but also reduced the treatment effect of gliomas by enhancing the levels of Treg cells, neutrophils, and M2-polarized subtype macrophages in the TIME to suppress antitumour immunity [121]. Tumor cells sacrificing themselves could also get the surrounding tumor cells to be in a stress state and finally avoid ferroptosis by secreting cytokines [145]. What substances are released by tumor cells after ferroptosis and the effects of these signals need to be further studied [153,154]. More evidence is needed to confirm whether the cytokines released by cells after ferroptosis enable surrounding glioma cells to evade immune surveillance by regulating TIME [155,156].

## 5. Conclusions and Future Perspectives

Ferroptosis, as a new programmed cell death mode, is different from other RCDs and is the result of iron-dependent lipid peroxidation accumulation. In this review, we focused on the regulatory mechanism of ferroptosis and found that ferroptosis plays multiple roles in the occurrence and development of glioma. Ferroptosis not only induces glioma cell death but also promotes glioma cell growth, invasion, migration, and resistance by regulating TAMs, MDSCs, Treg cells, neutrophils and CD8^+^ T cells in the TIME. Notably, studies have shown that some new compounds (such as strychnine, dihydroartemisinin and ibuprofen) are capable of inducing ferroptosis in gliomas, and ferroptosis-induced chemosensitizers, including erastin, can be used in combination with various drugs (such as cisplatin, temozolomide and cytarabine), which may provide new therapeutic opportunities for glioma treatment. However, poor BBB penetration reduces targeting tumor ability, and potential compensatory mechanisms hinder the effectiveness of ferroptosis agents in glioma therapy. It has been suggested that designing nanoengineered systems to improve the targeted delivery of drugs can overcome these issues, which further enhances the effectiveness of glioma treatment [157,158,159] (Table 2). Overall, although ferroptosis has great advantages in glioma treatment, we still need multidisciplinary cooperation to further explore the pros and cons of targeting ferroptosis and to evaluate its potential value in clinical applications.

## Figures and Tables

**Figure 1 antioxidants-11-02123-f001:**
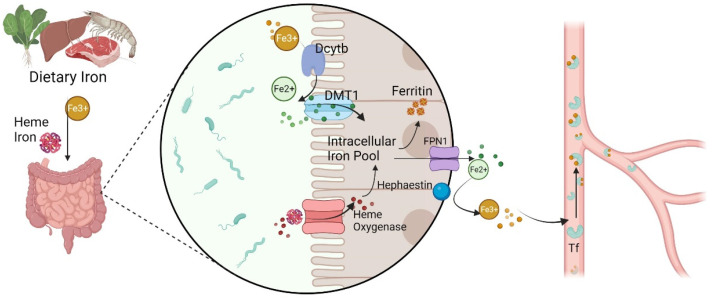
Iron absorption and metabolism in the body. Fe^2+^, ferrous cation; Fe^3+^, ferric cation; Dcytb, duodenal cytochrome b; DMT1, divalent metal transporter 1; FPN1, ferroportin 1.

**Figure 2 antioxidants-11-02123-f002:**
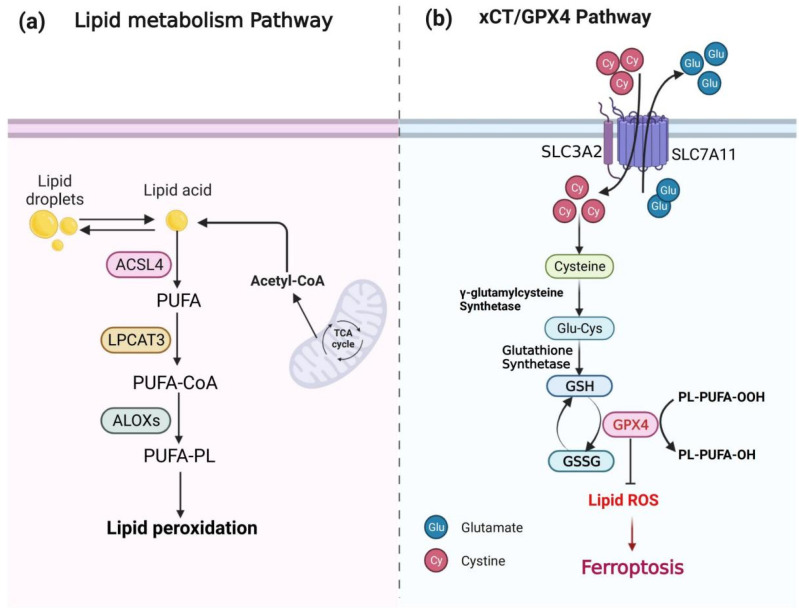
Molecular mechanisms of ferroptosis. (**a**) The lipid metabolism pathway; (**b**) the xCT/GPX4 pathway. ACSL4, long-chain fatty acid CoA ligase 4; PUFA, polyunsaturated fatty acid; LPCAT3, lysophosphatidylcholine acyltransferase 3; PUFA-CoA, polyunsaturated fatty acid coenzyme A; ALOXs, lipoxygenases; PUFA-PL, polyunsaturated fatty acid-containing phospholipid; TCA, tricarboxylic acid cycle; GSH, glutathione; GSSG, glutathione disulfide; GPX4, glutathione peroxidase 4.

**Figure 3 antioxidants-11-02123-f003:**
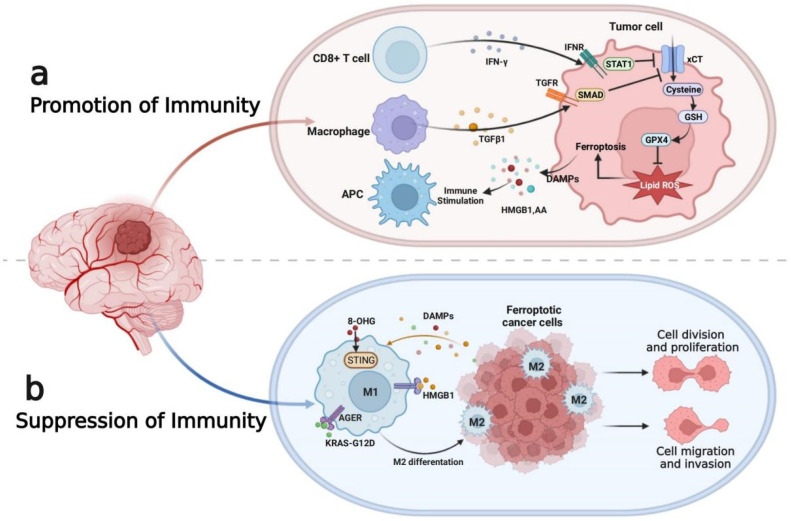
Role of ferroptosis in glioma immunity. (**a**) CD8^+^ T cells release IFNγ to activate INFR (which inhibits SLC7A11 transcription through STAT1) to promote tumour cell ferroptosis. TGFβ1 released by macrophages induces the downregulation of system xCT mediated by SMAD proteins, thereby triggering lipid ROS-mediated ferroptosis via the GSH-GPX4 axis. In turn, ferroptotic glioma cells release DAMPs (such as HMGB1 and AA) to promote the recruitment and activation of immune cells. (**b**) In contrast, DAMPs, such as HMGB1, KRAS-G12D and 8-OHG, could affect the function of macrophages in the tumour microenvironment. In particular, KRAS-G12D binds AGER on the cell surface of macrophages to trigger M2 macrophage polarization, which might limit antitumour immunity. IFN-γ, interferon-γ; INFR, interferon receptor; STAT1, signal transducer and activator of transcription 1; GSH, glutathione; GPX4, glutathione peroxidase 4; DAMPs, damage-associated molecular patterns; HMGB1, high mobility group protein B1; AA, arachidonic acid; 8-OHG, 8-hydroxyguanosine; AGER, advanced glycosylation end productspecific receptor.

**Table 1 antioxidants-11-02123-t001:** The features of different forms of RCD.

	Morphological Features		Biochemical Features	Common Inspection Indicators
Ferroptosis	cell membrane	plasma membrane integrity	Iron accumulation and lipid peroxidation	GSH, GPX4, MDA, SLC7A11, NRF2, ACSL4, FSP1,LPO
	Cell cytoplasm	small mitochondria and increased mitochondrial membrane densities		
	Cell nucleus	no obvious alteration		
Apoptosis	Cell membrane	plasma membrane disruption,	DNA fragmentation	Caspase, Bcl-2, TUNEL,Annexin-V JC-1
	Cell cytoplasm	cell volume reduction		
	Cell nucleus	nuclear volume reduction chromatin agglutination		
Necroptosis	Cell membrane	plasma membrane disruption,	Drop in ATP levels	RIP1, RIP3 Calcein-AM
	Cell cytoplasm	generalized swelling of the cytoplasm and organelles		
	Cell nucleus	oderate chromatin condensation and leakage of cellular constituents		
Autophagy	Cell membrane	no obvious alteration	Increased lysosomal activity	LC3, ATG family proteins(ATG5, ATG7)
	Cell cytoplasm	formation of double-membraned autolysosomes		
	Cell nucleus	no chromatin agglutination		

**Table 2 antioxidants-11-02123-t002:** Various NP-based drug delivery systems for the potential treatment of glioma.

Nanocarrier	Coating	Outcome	Reference
Cisplatin-Fe_3_O_4_ /Gd_2_O_3_	LF + RGD dimer	Increased accumulation in tumor Released Fe^2+^ and Fe^3+^	[160]
Iron oxide	NIR-fluorescent silica	Visualized tumor-associated macrophage populations	[161]
PEG	Doxorubicin	Increased drug diffusion across BBB	[162]
PEtOz-SS-PCL micelle	Doxorubicin	Increased drug diffusion across BBB	[163]
Liposome	Temozolomide	Enhanced antitumor activity	[164]
OX26-PLGA	Temozolomide	Enhanced permeability Improved cellular uptake	[165]
Fa-PEG-PCL	Luteolin	Prolonged survival time Enhanced antitumor activity	[166]
Anti-miR-21-PLA	Temozolomide	Increased apoptotic cell death	[167]
Transferrin-PEG-PLA	Resveratrol	Improved drug accumulation	[168]
Albumin	Paclitaxel and fenretinide	Increased drug diffusion across BBB Increased survival rate	[169]

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
