# Peer review of "Ferroptosis and Its Potential Role in Glioma: From Molecular Mechanisms to Therapeutic Opportunities"

_antioxidants, 2022, doi:10.3390/antiox11112123_

Round 1
Reviewer 1 Report
In this paper the authors present an overview of the potential role of ferroptosis in glioma. The manuscript is interesting and well-organized; however, there are some points that should be addressed by the authors:
in par. 2.1 lines 69-72, ferroportin should be mentioned as the iron exporter and refs 21,22 are not the most appropriate. PCBP2 is the intracellular iron chaperone that provides Fe2+ to ferroportin, it is not a ferroxidase as ceruloplasmin and hephaestin. Please correct!
line 80, the term 'recombinant' appears to be unnecessary.
line 118, replace 'steady' with 'stable'
lines 143-145, the sentence seems to be truncated
line 189, 'produced' or 'induced'?
lines 197, 218 and 230, 'encourages' or 'stimulates/promotes'?
paragraph 4 should be checked by a native English-speaker because various sentences are confusing (e.g., lines 341-345, 362-365, 374-378).
Reviewer 2 Report
To start with positives, the topic of the review, potential role of ferroptosis in glioma, is timely and important. The authors do cover a large range of relevant literature and subtopics, and have provided some high quality diagrams.
However, there are key shortcomings of the review, which would require an extensive overhaul of the manuscript:
Writing and logic flow overall is extremely hard to follow, even for readers versed in ferroptosis (which the review is not meant for). For example, lines 106-108: “Meanwhile, ACSL4 plays an irreplaceable role… for the biosynthesis of polyunsaturated fatty acids, which are used to develop(?) lipid peroxidation and ferroptosis” or line 118 “as a companion protein, helps xCT remain steady(?)”.; line 151-152: “Glioma cells that were administered propofol presented a lower glioma cell count”. As exemplified here, many sentences, while grammatically correct, are imprecise and it is unclear what the authors are trying to state. The manuscript suffers from somewhat poor writing throughout.
Furthermore, as exemplified in the paragraphs starting in line 178 and in line 206, facts/findings from the literature are just laid out one after another like a laundry list with no explanation and no transfer of useful knowledge to the reader. Simply put, it is impossible to follow and understand what the authors are trying to say. The writing throughout Section 3 would need a major overhaul. Using the line 206 paragraph as example: what is meant by managing cellular metabolism? What is meant by p53 reactivator? In sentence 211, why does increasing cysteine decrease NADPH? This is likely an error and the authors meant increase NADPH, but even if so, how? This needs to be explained. The next sentence authors state that these results are significantly different from conclusion of Takeuchi et al, but do not even state what Takeuchi’s conclusions were.
Additionally, there are not enough insights that pertain specifically to glioma, which is the key point of this review, considering that there are numerous other reviews available that explain ferroptosis. For example, insights in section “2. Molecular mechanisms of ferroptosis” are covered in many reviews, which itself is not a problem. However, once we get to section 3, “role of ferroptosis in glioma”, the same key points (why ACLS4 is important; why xCT is important; why GPX4 is important) are needlessly restated, and then a set of facts listed in an order that does not seem logical or well explained, as described above. Even at the expense of stating fewer facts, those facts that are mentioned need to be explained in greater depth.
Minor points:
The first few paragraphs of “Section 4 Challenges of ferroptosis in glioma” are all about ferroptosis and the tumor immune microenvironment and should have their own separate section.
The role of ubiquinol, as recently highlighted by findings on FSP1 and DHODH in ferroptosis, have been skipped and should be discussed in the review, including potential relevance to glioma / glioma treatment.
Prior to Stockwell groups findings that consolidated the term ferroptosis, there were key findings of iron dependent cell death (https://pubmed.ncbi.nlm.nih.gov/6086389/) and ros dependent cell death by cysteine depletion (https://pubmed.ncbi.nlm.nih.gov/11895126/) that laid prior groundwork, and this should be discussed.
Line 60: Ferroptosis differs from other major forms of cell death in what way?
Section 2.1: Some aspects of iron delivery to the brain as opposed to other organs should be discussed. Similarly, in 3.1, iron metabolism in glioma is only compared to that of normal brain or other brain tumors, but there should be a more thorough explanation about what is unique about the iron metabolism in gliomas as opposed to the prototypical non-brain solid tumor.
Reviewer 3 Report
In the review article titled "Ferroptosis and Its Potential Role in Glioma: from Molecular 2 Mechanisms to Therapeutic Opportunities," by Luo et al., the author discussed the mechanisms of ferroptosis in glioma. Ferroptosis is a recently defined cell death pathway, and I believe this is a timely update to the literature. I think the article is generally well-prepared. I have a couple of minor comments as follows:
1. In the section on iron metabolism (line 139), the author mainly discussed iron uptake from the microenvironment. I would like to point out that several recent studies indicated that iron homeostasis and ferroptosis are also affected by iron-sulfur cluster proteins (PMID: 34039609), which may be worthy of discussion in this review.
2. Most of the ferroptotic pathways are affected by a transcriptional factor, Nuclear factor erythroid 2-related factor 2 (NRF2), which might be worth mentioning here.
3. The author discussed ferroptosis in glioma, which includes both glioblastoma (GBM, WHO IV), IDH-mutated high-grade glioma (WHO IV), and lower-grade glioma (astrocytoma and oligodendroglioma, WHO II/III). However, the article primarily focused on IDH wild-type GBM but less on the other two disease subtypes. There are several recent articles indicate that these tumors are vulnerable to oxidative stress, lipid peroxidation, and ferroptosis as well (PMID: 32312817, PMID: 31548295, and PMID: 34250481). The author may include these latest findings.
4. I feel the articles minimally discussed the immune microenvironment. I think this may be removed from the abstract.
Round 2
Reviewer 1 Report
the authors have greatly improved the paper
Reviewer 2 Report
Authors have satisfactorily addressed my concerns.
Reviewer 3 Report
The author made improvements to the manuscript.